# Multi-period uncertain portfolio selection model with prospect utility function

**Gaohuizi Guo** ID *, **Yao Xiao, Cuiyou Yao**

School of Management and Engineering, Capital University of Economics and Business, Beijing, China

* huizi@cueb.edu.cn

## Abstract

In this paper, we discuss a multi-period portfolio optimization problem based on uncertainty theory and prospect theory. We propose an uncertain multi-period portfolio selection model, in which the return utility and risk of investment are measured by prospect theory utility function and uncertain semivariance. More realistically, the influence of transaction costs and bankruptcy of investor are also considered. Moreover, to solve the portfolio model, this paper designs a new artificial bee colony algorithm by combining sine cosine search method. Finally, a numerical experiment is presented to demonstrate the proposed model and the effectiveness of the designed algorithm.

## Introduction

Portfolio selection discusses the problem of how to allocate a certain amount of investors' wealth among different assets and form a satisfying portfolio. Markowitz [1] developed the classical mean-variance (M-V) model, which laid the foundation of modern portfolio analysis. By quantifying the investment return as expected value and quantifying the investment risk as variance, the investors seek a portfolio to strike a balance between maximizing the return and minimizing the risk. The conventional M-V model, on the other hand, is a single-period model that makes a single choice at the beginning of the period and adheres to it until the conclusion of the period. As we know, in real financial market, investor needs to timely adjust the investment decision according to financial market environment changes. Thus, expanding portfolio selection from a single time to many periods is essential. Numerous scholars have explored multiperiod portfolio selection issues so far. For example, Sun [2] shows that multi-period portfolio problems can be accomplished, essentially by means of a dynamic programming approach. Liu [3] proposed a hybrid particle swarm optimization algorithm to solve the multi-period portfolio selection model. Calafiore The model provided by [4] determines multi-period optimum portfolio changes with the goal of reducing a cumulative risk measure across the investment horizon. More literatures are as follows [5–7].

Using the framework of probability theory, the preceding portfolio models represent the returns of securities as random variables. The fundamental assumption among them is that security market conditions in the future may be accurately mirrored by security data from the past. However, in real life, due to the fact that there are often unexpected events in the securities market, historical data cannot represent future trends, and there are many new securities

**Funding:** This work was supported by Beijing Municipal Natural Science Foundation (No.9192005) and the special fund of basic

scientific research business fees of Beijing Municipal University of Capital University of Economics and Business (No.ZD202105). The funders are Prof. Cuiyou Yao, who is the project leader and whose main role in the research is the review and revision of the paper.

**Competing interests:** The authors have declared that no competing interests exist.

that have no historical data as a reference. In this situation, security returns can be obtained by the experts' subjective estimations instead of historical data. At present, there are two approaches to deal with the experts' subjective estimations: first, fuzzy set theory, proposed by Zadeh [8], and second, uncertainty theory developed by Liu [9]. However, existing studies have shown that, contradictions will arise if fuzzy variables are used to describe the security returns. For instance, we should have a membership function to describe a fuzzy variable, such as the security return rate. Assume it is a triangular fuzzy variable $\xi = (-0.2, 0.4, 1.0)$. From the membership function, we know from possibility theory (or credibility theory) that Pos $\xi = 0.4 = 1$ (or Cr $\xi = 0.4 = 0.5$), which indicates that the return rate is precisely 0.4 with a belief degree of 1 in the possibility measure (or 0.5 in credibility measure). This conclusion is illogical, though. Liu suggested an uncertain measure and unknown variable to address these issues. (see Liu [10], [11], Qin [12]).

In the meantime, uncertain portfolio selection problems also have been widely concerned in two directions. The former direction focused on single-period portfolio selections. For example, Huang [13, 14] demonstrates that uncertain variable should be reflected upon the experts' individualized assessments of security returns. Torre et al. [15] proposed a stock portfolio selection model with set-valued analysis in an uncertain environment. Zhang [16] explores single-period portfolio selection problem in uncertain environment in which security returns cannot be well reflected by historical data, but can be assessed by the experts. More literatures are as follows [13, 17, 18]. While, the latter direction concentrated on multi-period portfolio selections. Zhou [19] studied a time consistent multi-period rolling portfolio optimization problem under fuzzy environment. Li et al. [20] examined an uncertain multi-period portfolio selection problem in three steps with the influence of transaction cost and bankruptcy of investor being considered. Chen et al. [21] presented a novel uncertain multi-period multi-objective mean-variance-skewness model by taking into account multiple realistic investment constraints, such as transaction cost, bounds on holdings, and cardinality etc. To our knowledge, except for the researches mentioned hereinbefore, uncertainty theory has been rarely used for multi-period portfolio selections. The lack of work and the growing interest of the scientific community in uncertainty theory has motivated this work.

Markowitz's portfolio theory is based on the rationality of investors. However, investors in real life are not always completely rational when making investment decisions. In most cases, people are bounded rational, they may incorporate their preferences when making decisions. Their decisions may be based more on subjective factors. The widely accepted theory of the irrational behavior of investors is the Prospect Theory proposed by Kahnemann and Tversky [22, 23]. The theory effectively combines psychology and economics through experimental research and studies various types of irrational decision-making behaviors. Shefrin and Statman created a behavioral portfolio theory based on the principles of SP/A theory and prospect theory. There is a substantial body of literature on behavioral portfolio theory since the groundbreaking work of Shefrin and Statman [24]. In order to capture investors' behavioral characteristics in defined contribution pension planning, Blake et al. [25] also developed an asset allocation model that makes use of the prospect theory value function.

The objective of this paper is to discuss a multi-period portfolio selection problem under assumption that the security returns are assumed as uncertain variables. The main contributions of this work can be summarized as follows: (1) This paper proposes an uncertain multi-period portfolio optimization model, in which the utility of investment returns is measured by a utility function in prospect theory. (2) The investment risk is measured by uncertain semi-variance instead of uncertain variance. Also, the proposed model simultaneously considers transaction costs and bankruptcy constraint. (3) This paper proposes a new artificial bee

colony algorithm by combining sine cosine search method to improve the efficiency and precision of solving the portfolio model.

The remainder of the paper is structured as follows: In Section 2, we cover fundamental uncertainty theory ideas and findings. In Section 3, we provide a multi-period utility-semi-variance model for portfolio selection under uncertainty. In Section 4, we provide a modified method for an artificial bee colony to solve the given problem. The section concludes with an illustration of the usefulness of the Section 5 recommendations. Section 6 gives conclusions.

## Preliminaries

The following introduces some fundamental concepts related to uncertainty theory.

**Definition 1** [9]. Assume that $\Gamma$ is a nonempty set and $\mathcal{Z}$ is a $\sigma$-algebra on $\Gamma$. If a set function $M$ is called an uncertain measure, then the following axioms hold:

Axiom 1: (Normality) $M\{\Gamma\} = 1$;

Axiom 2: (Duality) $M\{\Lambda\} + M\{\Lambda^c\} = 1$ for any event $\Lambda \in L$;

Axiom 3: (Subadditivity) For any given countable sequence $\Lambda_1, \Lambda_2, \ldots$, we have

$$M\left\{\bigcup_{i=1}^{\infty} \Lambda_i\right\} \leq \sum_{i=1}^{\infty} \{\Lambda_i\}.$$

Axiom 4: (Product) Assume that $(\Gamma_k, L_k, M_k)$ is an uncertainty space for all $k = 1, 2, \ldots, \infty$. Then, the product uncertain measure $M$ satisfies the following relation

$$M\left\{\prod_{k=1}^{\infty} \Lambda_k\right\} = \bigwedge_{k=1}^{\infty} M_k\{\Lambda_k\}.$$

Here, $\Lambda_K$ is randomly chosen events from $L_k$ for $k = 1, 2, \ldots$.

An uncertain variable $\xi$ is defined by Liu [9] as a measurable function from an uncertainty space $(\Gamma, \mathcal{L}, \mathcal{M})$ to the set of real numbers, i.e., for any Borel set B of real numbers, the set

$$\{\xi \in B\} = \{\gamma \in \Gamma \mid \xi(\gamma) \in B\}.$$

is an event.

**Lemma 1** [10]. Any uncertain measure $\mathcal{M}$ is increasing, i.e. for any events $\wedge_1 \subset \wedge_2$, we have

$$\mathcal{M}\{\wedge_1\} \leq \mathcal{M}\{\wedge_2\}.$$

**Definition 2** [9]. The uncertainty distribution $\Phi : \mathcal{R} \to [0, 1]$ of an uncertain variable $\xi$ is defined by

$$\Phi(t) = \mathcal{M}\{\xi \leq t\},$$

for any real number t.

**Definition 3** [9]. Let $\xi$ be an uncertain variable. Then the expected value of uncertain variable $\xi$ is defined by

$$E[\xi] = \int_0^{+\infty} \mathcal{M}\{\xi \geq r\}dr - \int_{-\infty}^0 \mathcal{M}\{\xi \leq r\}dr.$$

Provided that at least one of the two integrals is finite.

**Definition 4** [9]. An uncertainty distribution $\Phi(x)$ is said to be regular if it is a continuous and strictly increasing function with respect to $x$ at which $0 < \Phi(x) < 1$, and

$$\lim_{x \to -\infty} \Phi(x) = 0, \qquad \lim_{x \to +\infty} \Phi(x) = 1.$$

An uncertain variable $\xi$ is called zigzag if it has a zigzag uncertainty distribution

$$\Phi(\xi) = \begin{cases} 0, & \xi \le a, \\[2mm] \dfrac{\xi - a}{2(b - a)}, & a \le \xi \le b, \\[2mm] \dfrac{\xi + c - 2b}{2(c - b)}, & b \le \xi \le c, \\[2mm] 1, & \xi \ge c. \end{cases} \tag{1}$$

Defined by $\xi \sim \mathcal{Z}(a, b, c)$, where $a$, $b$ and $c$ are real numbers with $a < b < c$. The inverse uncertainty distribution of zigzag uncertain variable $\mathcal{Z}(a, b, c)$ is

$$\Phi^{-1}(\alpha) = \begin{cases} (1 - 2\alpha)a + 2\alpha b, & 0 < \alpha < 0.5, \\[2mm] (2 - 2\alpha)b + (2\alpha - 1)c, & 0.5 \le \alpha < 1. \end{cases} \tag{2}$$

**Lemma 2.** [9]. Let $\xi$ be an uncertain variable. Then for any given numbers $a > 0$ and $p > 0$, we have

$$\mathcal{M}\{|\xi| > a\} \le \frac{E[|\xi|^p]}{a^p}. \tag{3}$$

**Lemma 3.** [10]. Let $\xi$ be an uncertain variable with regular uncertainty distribution $\Phi$. If the expected value exists, then

$$E[\xi] = \int_0^1 \Phi^{-1}(\alpha)d\alpha. \tag{4}$$

**Definition 5** [14]. Let $\xi$ be an uncertain variable with finite expected value e. Then the semi-variance of $\xi$ is demonstrated by

$$SV[\xi] = E[[(\xi - e)^-]^2],$$

where

$$(\xi - e)^- = \begin{cases} \xi - e, & \text{if } \xi \le e, \\[2mm] 0, & \text{if } \xi > e. \end{cases}$$

When the uncertain variable $\xi$ has continuous uncertainty distribution $\Phi(r)$, consequently

$$
\begin{aligned}
SV[\xi] &= \int_0^{+\infty} \mathcal{M}\{((\xi - e)^-)^2 \geq r\} dr \\
&= \int_0^{+\infty} \mathcal{M}\{\xi \leq e - \sqrt{r}\} dr \\
&= \int_{-\infty}^e 2(e - r)\Phi(r) dr.
\end{aligned}
$$

## Multi-period uncertain portfolio model with utility function

In this part, we offer a multi-period uncertain portfolio model that uses the prospect theory utility function to describe investors' behavior elements in the multi-period portfolio by combining uncertainty theory and prospect theory. And the investment risk is measured by uncertain semivariance instead of uncertain variance. At the same time, we consider the portfolio model in the case of bankruptcy constraints. For the sake of clarity, we first define all of the notations that will be used in the following sections. For $i = 1, 2, \ldots, n$, and $t = 1, 2, \ldots, T$, we let

$r_{t,i}$, the uncertain return rate of $i$th risk security at period $t$;

$d_{t,i}$, the transaction cost of security $i$ at period $t$;

$W_t$, the available wealth at the end of period $t$;

$\varepsilon_{t,i}$, the lower bound of security $i$ at period $t$;

$\delta_{t,i}$, the upper bound of security $i$ at period $t$;

$m_t$, the maximum number of securities hold in the portfolio at period $t$, $1 \leq m_t \leq n$;

$z_{t,i}$, the binary variable, is 1 if $z_i = 1$, 0 otherwise.

### Dynamic prospect theory utility function

Prospect theory defines how investors assess assets through comparing them to a reference value, where results falling below the reference value are considered losses and results rising beyond the reference value are considered gains. Additionally, investors exhibit loss aversion, which is the tendency to be more sensitive to losses than profits. The incorporation of loss aversion into the multi-period portfolio model, which incorporates a crucial behavioral trait in investors' decision-making, is one of our key foci in this work. The utility function is expressed by weighting formula and value formula, which is expressed by

$$
V = \sum_{t=1}^T (\omega(p)v(E[R_{t,N}])). \tag{5}
$$

Three essential traits define the value function. (1) Reference dependence: People compare assets to a particular reference value while evaluating them. (2) Loss aversion: People are less tolerant of profits than losses. (3) Decreasing sensitivity: People are often risk-averse when it comes to profits and risk-seeking when it comes to losses. In this paper we will employ the

piecewise value function of Tversky and Kahneman [26], which can be formulated as

$$
v(E[R_{t,N}]) = \begin{cases} (E[R_{t,N}] - \theta)^{\alpha}, & E[R_{t,N}] \geq \theta, \\ -\lambda(\theta - E[R_{t,N}])^{\beta}, & E[R_{t,N}]] < \theta, \end{cases} \quad t = 1, \ldots, T \tag{6}
$$

where $\theta$ denotes the given reference return. $\lambda$ denotes the loss aversion ratio, and $\lambda > 1$, indicating loss aversion. $\alpha$ and $\beta$ denote the curvature parameters for gains and losses respectively, the greater the $\alpha$ and $\beta$ values, the greater the risk preference of investors. According to Tversky and Kahneman [23], $\lambda = 2.25$, $\alpha = \beta = 0.88$.

We will use Tversky and Kahneman's original probability weighting function [23], which can be expressed as,

$$
\omega(p) = \frac{p^{\gamma}}{(p^{\gamma} + (1-p)^{\gamma})^{\frac{1}{\gamma}}}, \tag{7}
$$

with $0 < \gamma \leq 1$. According to Tversky and Kahneman's experiments [23], the median value for *gamma* is 0.65.

The security returns $r_{t,i}$ ($i = 1, 2, \ldots, n$; $t = 1, 2, \ldots, T$) are supposed as zigzag uncertain variables, defined by $\mathcal{Z}(a_{t,i}, b_{t,i}, c_{t,i})$, where $a_{t,i}$, $b_{t,i}$ and $c_{t,i}$ are real numbers with $a_{t,i} < b_{t,i} < c_{t,i}$. Moreover, the transaction costs are considered. The net return of the portfolio $x_t$ at period $t$ is

$$
\begin{aligned}
R_{t,N} &= \sum_{i=1}^{n} r_{t,i} x_{t,i} - D_t \\
&= \sum_{i=1}^{n} (r_{t,i} x_{t,i} - d_{t,i} |x_{t,i} - x_{t-1,i}|).
\end{aligned} \tag{8}
$$

According to Liu [9], we can obtain the expected value of $r_{t,i}$ as follows,

$$
E(r_{t,i}) = \frac{a_{t,i} + 2b_{t,i} + c_{t,i}}{4}. \tag{9}
$$

Therefore, the expected value of net return is

$$
E[R_{t,N}] = \sum_{i=1}^{n} \left( \frac{a_{t,i} + 2b_{t,i} + c_{t,i}}{4} x_{t,i} - d_{t,i} |x_{t,i} - x_{t-1,i}| \right). \tag{10}
$$

## The cumulative risk

To overcome the shortcoming of variance risk measure, semivariance was proposed by Markowitz [27]. Semivariance more closely approximates investors' intuitive sense of risk than variance since it only evaluates the variability of returns below the mean and ignores any variability of returns above the mean. Therefore, in this paper, by considering the security returns as uncertain variables, an uncertain semivariance is introduced, and then its' crisp form is derived.

The risk of the portfolio $x_t = (x_{t,1}, x_{t,2}, \ldots, x_{t,n})$ at period $t$ is expressed as

$$
\begin{aligned}
SV[R_{t,N}] &= SV\left[\sum_{i=1}^{n}\left(r_{t,i}x_{t,i} - d_{t,i}|x_{t,i} - x_{t-1,i}|\right)\right] \\
&= SV\left[\sum_{i=1}^{n} x_{t,i}^2 r_{t,i}\right].
\end{aligned}
\tag{11}
$$

According to Definition 5, we can obtain

$$
SV[r_{t,i}] = \int_{-\infty}^{\frac{a_{t,i} + 2b_{t,i} + c_{t,i}}{4}} 2\left(\frac{a_{t,i} + 2b_{t,i} + c_{t,i}}{4} - r\right)\Phi(r)dr.
$$

Since $\Phi(r)$ is a piecewise function, we can obtain the following results from two cases: if $a_{t,i} + c_{t,i} \le 2b_{t,i}$, then

$$
\begin{aligned}
SV[r_{t,i}] &= \int_{a_{t,i}}^{\frac{a_{t,i} + 2b_{t,i} + c_{t,i}}{4}} 2\left(\frac{a_{t,i} + 2b_{t,i} + c_{t,i}}{4} - r\right)\left(\frac{r - a_{t,i}}{2(b_{t,i} - a_{t,i})}\right)dr \\
&= \frac{(2b_{t,i} + c_{t,i} - 3a_{t,i})^3}{384(b_{t,i} - a_{t,i})}.
\end{aligned}
\tag{12}
$$

If $a_{t,i} + c_{t,i} > 2b_{t,i}$, then

$$
\begin{aligned}
SV[r_{t,i}] &= \int_{a_{t,i}}^{b_{t,i}} 2\left(\frac{a_{t,i} + 2b_{t,i} + c_{t,i}}{4} - r\right)\left(\frac{r - a_{t,i}}{2(b_{t,i} - a_{t,i})}\right)dr \\
&\quad + \int_{b_{t,i}}^{\frac{a_{t,i} + 2b_{t,i} + c_{t,i}}{4}} 2\left(\frac{a_{t,i} + 2b_{t,i} + c_{t,i}}{4} - r\right)\left(\frac{r + c_{t,i} - 2b_{t,i}}{2(c_{t,i} - b_{t,i})}\right)dr \\
&= \frac{1}{24}(a_{t,i} - b_{t,i})(a_{t,i} + 2b_{t,i} - 3c_{t,i}) + \frac{(a_{t,i} - 2b_{t,i} + c_{t,i})^2(a_{t,i} - 14b_{t,i} + 13c_{t,i})}{384(c_{t,i} - b_{t,i})}.
\end{aligned}
\tag{13}
$$

Furthermore, the cumulative investment risk over $T$ period is expressed as follows:

$$
\sum_{t=1}^{T} SV[R_{t,N}] = 
\begin{cases}
\sum_{t=1}^{T}\sum_{i=1}^{n} x_{t,i}^2 \dfrac{(2b_{t,i} + c_{t,i} - 3a_{t,i})^3}{384(b_{t,i} - a_{t,i})}, & a_{t,i} + c_{t,i} \le 2b_{t,i}, \\[2ex]
\sum_{t=1}^{T}\sum_{i=1}^{n} x_{t,i}^2 \left[\dfrac{1}{24}(a_{t,i} - b_{t,i})(a_{t,i} + 2b_{t,i} - 3c_{t,i})\right. \\[1ex]
\left. + \dfrac{(a_{t,i} - 2b_{t,i} + c_{t,i})^2(a_{t,i} - 14b_{t,i} + 13c_{t,i})}{384(c_{t,i} - b_{t,i})}\right], & a_{t,i} + c_{t,i} > 2b_{t,i}.
\end{cases}
\tag{14}
$$

## Bankruptcy constraint

The occurrence of bankruptcy refers to the investor's wealth below a certain value during the investment period or the end of the investment. The value is called the bankruptcy level, $\beta_t$ is

used to indicate the bankruptcy level at time $t$, and $BR_t$ is used to indicate the bankruptcy event at time $t$. For the portfolio, the belief degree of bankruptcy at time $t$ is:

$$\mathcal{M}\{BR_t\} = \mathcal{M}\{W_t < \beta_t, W_s > \beta_s, s = 1, 2, \ldots, t-1\}, t = 1, 2, \ldots, T. \tag{15}$$

**Proposition 1.** Assume that $E[W_t] > \beta_t$, and $\frac{E[((W_t - E[W_t])^-)^2]}{[E[W_t] - \beta_t]^2} \le \varepsilon_t$. Then $\mathcal{M}\{BR_t\} \le \varepsilon_t$.

**Proof.** Let $|\xi| = |(W_t - E[W_t])^-|$, $\mu = E[W_t] - \beta_t$ and $p = 2$ in Lemma 2. Since $E[W_t] > \beta_t$ and $\frac{E[\,|(W_t - E[W_t])^-|\,]}{E[W_t] - \beta_t} \le \varepsilon_t$, the following inequality holds

$$\mathcal{M}\{\,|(W_t - E[W_t])^-| \ge E[W_t] - \beta_t\} \le \frac{E[((W_t - E[W_t])^-)^2]}{[E[W_t] - \beta_t]^2} \le \varepsilon_t. \tag{16}$$

Therefore,

$$\begin{aligned}
\mathcal{M}\{BR_t\} &= \mathcal{M}\{W_t < \beta_t, W_s > \beta_s, s = 1, 2, \ldots, t-1\} \\
&\le \mathcal{M}\{W_t < \beta_t\} \\
&\le \mathcal{M}\{\,|(W_t - E(W_t))^-| \ge E(W_t - b_t)\} \\
&\le \frac{E[((W_t - E[W_t])^-)^2]}{[E[W_t] - \beta_t]^2} \le \varepsilon_t.
\end{aligned} \tag{17}$$

The proof is completed.

The wealth at the end of period $t$ is given by

$$\begin{aligned}
W_t &= W_{t-1}(1 + R_{t,N}) \\
&= W_{t-1}\left(1 + \sum_{i=1}^{n}\left(r_{t,i}x_{t,i} - d_{t,i}|x_{t,i} - x_{t-1,i}|\right)\right).
\end{aligned} \tag{18}$$

Assume that the initial wealth is $W_0$, solving Eq (18), we can get the terminal wealth at the end of period T,

$$W_T = W_0\prod_{t=1}^{T}\left(1 + \sum_{i=1}^{n}\left(r_{t,i}x_{t,i} - d_{t,i}|x_{t,i} - x_{t-1,i}|\right)\right). \tag{19}$$

Thus, the expected value of the terminal wealth can be expressed as

$$\begin{aligned}
E[W_T] &= W_0\prod_{t=1}^{T}\left[1 + \sum_{i=1}^{n}\left(x_{t,i}E(r_{t,i}) - d_{t,i}|x_{t,i} - x_{(t-1),i}|\right)\right] \\
&= W_0\prod_{t=1}^{T}\left[1 + \frac{1}{4}\left(\sum_{i=1}^{n}a_{t,i}x_{t,i} + 2\sum_{i=1}^{n}b_{t,i}x_{t,i} + \sum_{i=1}^{n}c_{t,i}x_{t,i}\right) - \sum_{i=1}^{n}d_{t,i}|x_{t,i} - x_{(t-1),i}|\right].
\end{aligned} \tag{20}$$

From the Eq (19), we can obtain

$$SV[W_T] = W_0^2\prod_{t=1}^{T}SV[R_{t,N}]. \tag{21}$$

In this paper, we let the expected wealth and semivariance wealth at period $t$ to be equal to the expected value of the terminal wealth at $T = t$, that is,

$$E[W_t] = W_0 \prod_{k=1}^{t} \left[ 1 + \frac{1}{4} \left( \sum_{i=1}^{n} a_{k,i} x_{k,i} + 2 \sum_{i=1}^{n} b_{k,i} x_{k,i} + \sum_{i=1}^{n} c_{k,i} x_{k,i} \right) - \sum_{i=1}^{n} d_{k,i} |x_{k,i} - x_{(k-1),i}| \right]. \quad (22)$$

$$SV[W_t] = W_0^2 \prod_{k=1}^{t} SV[R_{k,N}]. \quad (23)$$

Based on the above, the Eq (17) is equal to

$$\frac{E[((W_t - E[W_t])^-)^2]}{[E[W_t] - \beta_t]^2} = \frac{SV[W_t]}{[E[W_t] - \beta_t]^2} \leq \varepsilon_t. \quad (24)$$

## Model formulation

On the basis of the above analysis, an investor's investing objective is to maximize the prospect theory utility value and minimize the semivariance of their portfolio. Following the preceding description, the multi-period portfolio model may be expressed as follows:

$$\begin{cases} \max & \sum_{t=1}^{T} (\omega(p) v(E[R_{t,N}])) \\ \\ \text{s.t.} & \sum_{t=1}^{T} SV[R_{t,N}] \leq \alpha, & (25a) \\ \\ & \frac{E[((W_t - E[W_t])^-)^2]}{[E[W_t] - \beta_t]^2} \leq \varepsilon_t, & (25b) \\ \\ & \sum_{i=1}^{n} x_{t,i} = 1, & (25c) \\ \\ & i = 1, 2, \ldots, n, \ t = 1, 2, \ldots, T. \end{cases} \quad (25)$$

Where $\alpha$ is the maximum risk tolerance level, Eq (25a) confirms the semivariance of the portfolio can not surpass the specified minimal risk $\alpha$ at each period. Eq (25b) ensures that the sum of the weight associated with each security is equal to one, i.e., all the available money is invested in one portfolio. Eq (25c) represents the desired number of securities in the portfolio must not surpass the given value $m_t$ at each period.

Then, the model (25) can be equivalently translated into the following portfolio selection problems.

1. If $a_{t,i} + c_{t,i} \leq 2b_{t,i}$, the model (25) equals

$$
\begin{cases}
\max \quad \sum_{t=1}^{T}(\omega(p)v(E[R_{t,N}])) \\[2ex]
\text{s.t.} \quad \sum_{t=1}^{T}\sum_{i=1}^{n}x_{t,i}^2 \frac{(2b_{t,i}+c_{t,i}-3a_{t,i})^3}{384(b_{t,i}-a_{t,i})} \leq \alpha, \\[2ex]
\qquad \frac{W_0^2 \prod_{k=1}^{t} SV[R_{k,N}]}{[E[W_t]-\beta_t]^2} \leq \varepsilon_t, \\[2ex]
\qquad \sum_{i=1}^{n}x_{t,i}=1, \\[2ex]
\qquad i=1,2,\ldots,n, \ t=1,2,\ldots,T.
\end{cases}
\tag{26}
$$

2. If $a_{t,i} + c_{t,i} \geq 2b_{t,i}$, the model (25) equals

$$
\begin{cases}
\max \quad \sum_{t=1}^{T}(\omega(p)v(E[R_{t,N}])) \\[2ex]
\text{s.t.} \quad \sum_{t=1}^{T}\sum_{i=1}^{n}x_{t,i}^2 \Bigg[ \frac{1}{24}(a_{t,i}-b_{t,i})(a_{t,i}+2b_{t,i}-3c_{t,i}) \\[2ex]
\qquad + \frac{(a_{t,i}-2b_{t,i}+c_{t,i})^2(a_{t,i}-14b_{t,i}+13c_{t,i})}{384(c_{t,i}-b_{t,i})} \Bigg] \leq \alpha, \\[2ex]
\qquad \frac{W_0^2 \prod_{k=1}^{t} SV[R_{k,N}]}{[E[W_t]-\beta_t]^2} \leq \varepsilon_t, \\[2ex]
\qquad \sum_{i=1}^{n}x_{t,i}=1, \\[2ex]
\qquad i=1,2,\ldots,n, \ t=1,2,\ldots,T.
\end{cases}
\tag{27}
$$

## Solution algorithm

Notably, the model (25) presented in the preceding section is a multi-period optimization issue for which conventional optimization approaches may not provide the best solution. Therefore, we design a new artificial bee colony algorithm hybrid sine cosine search method to solve the portfolio model effectively.

The artificial bee colony (ABC) algorithm, which includes three types of bees: employed workers, observer bees, and scouts, was first put out by [28]. Equal numbers of bees are employed and observers, which together make up half of the colony. A food source's location offers a potential solution to the optimization issue, and the quantity of nectar it produces indicates the appropriate fitness value. It should be remembered that only one hired bee has access to a given food source. ABC has been compared favorably to other evolutionary algorithms due to its basic structure and straightforward implementation [29]. In addition, many heuristic

techniques often used a randomization procedure to perturb a given solution ordered to create a diversified range of solutions. If these answers are not diverse, the solution process will get stale very quickly. On the other hand, too diverse answers would lead to a fully random search process. The ABC heuristic is an example of an approach that combines diversity with intensification.

Thus, many issues in the real world have been addressed [30]. For instance, multi-objective optimization problems [31], binary optimization problems [32], data clustering problems [33]. Especially, ABC algorithm is widely used in portfolio selection. For portfolio selection with cardinality constraints and investment limit constraints, Kumar and Mishra [34] proposed a potent co-variance guided ABC. Kalayci et al. [35] proposed an enhanced technique built on the ABC algorithm, showcasing its better performance on benchmark data sets with procedures that effectively manage boundary constraints and cardinality constraints as well as processes that enforce feasibility and tolerate infeasibility. Therefore, this paper improves the ABC algorithm to solve the multi-period portfolio problems.

However, the ABC method will ultimately have poor convergence since the search equation performs better in exploration but worse in exploitation. To remedy the issue, researchers have created a plethora of diverse strategies. For example, Zeng et al. [36] suggested an efficient ABC based on an adaptive search method and a random grouping mechanism. Liu et al. [37] introduced an improved ABC that enhances the exploration ability by modifying the behaviors of bees. Bayraktar et al. [38] improved the exploitation and exploration ability of the ABC by ultilizing memory mechanisms and genetic operators to produce three imporved ABC algorithms. Singh and Sundar [39] employed two neighbourhood strategies that help ABC algorithm in faster convergence towards finding high quality solutions. Besides this, there are other studies [40, 41]. In this paper, to solve the issue of poor performance of the search equation in exploitation, we propose a new artificial bee colony algorithm by combining sine cosine search method, in which the population of the employed bee be changed to balance between the global and local searches.

Presuming that the initial population, consisting of *SN* solutions with D-dimensional vector $X_i = (x_{i,1}, x_{i,2}, \ldots, x_{i,D})$ is randomly produced as follows:

$$x_{i,j} = x_j^{min} + rand(0, 1)(x_j^{max} - x_j^{min}), \tag{28}$$

where $i \in \{1, 2, \ldots, SN\}, j \in \{1, 2, \ldots, D\}$. $x_j^{min}$ and $x_j^{max}$ demonstrate the lower and upper bounds of *j*th dimension, respectively. rand(0,1) demonstrates a random parameter uniformly distributed in (0,1).

As in employed bee period, bee motions are determined by the original ABC's randomness. This may impact the balance between the global and local searches. Therefore, in this paper, we changed the population of the employed bee into two, one population uses the original formula (29) to search for new solutions, and the other population searches for new solutions according to Eq (30). Then compare the new solutions generated by the two methods, leaving the better solution.

$$v_{i,j} = x_{i,j} + \phi_{i,j}(x_{i,j} - x_{k,j}), \tag{29}$$

$$v_{i,j} = \begin{cases} x_{i,j} + r_1 \sin(r_2)|r_3 x_i^{max} - x_{i,j}|, & if \quad r_4 < 0.5, \\ x_{i,j} + r_1 \cos(r_2)|r_3 x_i^{max} - x_{i,j}|, & if \quad r_4 \geq 0.5. \end{cases} \tag{30}$$

In Eq (29), $i \in \{1, 2, \ldots, SN\}$ and $j \in \{1, 2, \ldots, D\}$ are randomly selected indexes; $\phi_{i,j}$ is a uniformly random value in [-1, 1].

In Eq (30), the parameter $r_1$ is a random vector that may determine the region of the search space surrounding the current solution. In addition to aiding in the exploration and exploitation of a search space, this parameter encourages a good balance between the two processes. The vector $r_1$ can be defined $r_1 = a - t\frac{a}{T}$ ($t$ is the current iteration, $T$ is the maximum number of iterations, and a is a constant.) The parameter $r_2$ is a random number in [0,2$\pi$], which is used to decide the direction of a current solution. The parameter $r_3$ is a random number in [0, 2], which provides a weight to $x_i^{max}$, it emphasizes the exploration ($r_3 > 1$) and exploitation ($r_3 < 1$). $r_4$ is a random number in the range [0, 1] that facilitates the transformation from sine to cosine operations and conversely.

Then, a greedy selection method based on nectar amount quality is used to choose the superior option between the candidate solution and the original solution. Then, the hired bees use dance to alert observer bees about food supplies.

While observers update similarly to employed bees throughout this phase, the primary distinction between them is that observers choose possible food sources to exploit based on probabilities determined by fitness values. The probability $p_i$ and fitness value $fit_i$ of the solution $X_i$ are computed as follows, assuming a minimization problem:

$$p_i = \frac{fit_i}{\sum_{i=1}^{SN} fit_i}, \tag{31}$$

$$fit_i = \begin{cases} \dfrac{1}{1+f_i}, & if \quad f_i \geq 0, \\ f_i, & if \quad f_i < 0, \end{cases} \tag{32}$$

where $f_i$ denotes the objective function value.

During the scout phase, in the original ABC algorithm, the scout bee searches for a new food source through the Eq (29). The method's neighborhood search does not have the global ideal value, as can be observed from the search formula. As a result, the algorithm has poor searching capabilities close to the global optimal value, which causes the algorithm to converge slowly. Therefore, in the modified ABC algorithm, the scout bee searches for the new food source by the following:

$$x_{i,j}^{new} = x_{i,j} + \phi_{i,j}(x_j^{max} - x_{i,j}) - \psi_{i,j}(x_j^{min} - x_{i,j}), \tag{33}$$

where $\phi_{i,j}$ and $\psi_{i,j}$ are two different random numbers in the range [0, 1]. The term $\phi_{i,j}(x_j^{max} - x_{i,j})$ expoeses the tendency of the solution to move toward the best solution, while the term $-\psi_{i,j}(x_j^{min} - x_{i,j})$ exposes the tendency of the solution to avoid the worst possible solution.

Given the above, we utilize ABC algorithm to address the two models (26) and (27), where the processes of the ABC algorithm are outlined as following:

**Step 1:** Set parameters: swarm size *pop_size*, boundaries of variables *bound* and maximum generation number *MAXGEN*;

**Step 2:** Using the constraint-handling techniques, randomly produce N initial people and turn them into matching feasible persons;

**Step 3:** Calculate the fitness levels of each people;

**Step 4:** Perform employed bee phase by Eqs (29) and (30);

**Step 5:** Select potential food sources by Eq (31), compare the fitness values of the offspring by Eq (32);

**Table 1. The zigzag uncertain returns of 10 securities.**

| T | Security $i$ | 1 | 2 | 3 | 4 | 5 | 6 | 7 | 8 | 9 | 10 |
|---|---|---|---|---|---|---|---|---|---|---|---|
| $t = 1$ | $a_{1,i}$ | -0.4 | -0.8 | -0.5 | -0.7 | -0.6 | -0.3 | -0.8 | -0.5 | -0.7 | -0.6 |
| | $b_{1,i}$ | 0.1 | -0.2 | -0.1 | 0.1 | 0.3 | -0.1 | 0.1 | 0.3 | 0.2 | 0.002 |
| | $c_{1,i}$ | 0.5 | 0.2 | 0.3 | 0.6 | 0.9 | 0.4 | 0.9 | 0.7 | 0.6 | 0.8 |
| $t = 2$ | $a_{2,i}$ | -0.3 | -0.7 | -0.5 | -0.6 | -0.5 | -0.5 | -0.8 | -0.5 | -0.7 | -0.5 |
| | $b_{2,i}$ | 0.2 | -0.1 | -0.2 | 0.1 | 0.3 | 0.1 | 0.1 | 0.3 | 0.2 | 0.2 |
| | $c_{2,i}$ | 0.5 | 0.3 | 0.5 | 0.5 | 0.8 | 0.4 | 0.9 | 0.7 | 0.6 | 0.9 |
| $t = 3$ | $a_{3,i}$ | -0.4 | -0.8 | -0.5 | -0.8 | -0.6 | -0.6 | -0.9 | -0.5 | -0.6 | -0.6 |
| | $b_{3,i}$ | 0.1 | -0.2 | -0.1 | 0.2 | 0.4 | -0.2 | -0.1 | 0.3 | 0.3 | 0.003 |
| | $c_{3,i}$ | 0.5 | 0.2 | 0.3 | 0.7 | 0.7 | 0.7 | 0.8 | 0.6 | 0.7 | 0.8 |

**Step 6:** Perform scout phase by Eq (33);

**Step 7:** Verify the stopping standard. If the stopping requirement is met, the iteration process should be terminated, and the best person should be reported as the best solution. Otherwise, go back to Step 3.

## Numerical example

In this part, we demonstrate the suggested methods using a numerical example. In this example, we suppose that the investor wishes to invest in the 10 securities for three consecutive periods, that is, $N = 10$ and $T = 3$. Here, the initial wealth $W_0 = 10,000$ RMB. And we assume that the return rates are characterized by zigzag uncertain variables with $r_{t,i} \sim \mathcal{Z}(a_{t,i}, b_{t,i}, c_{t,i})$ $(t = 1, 2, 3, i = 1, 2, \ldots, 10)$, as shown in Table 1.

In addition, we assume that the transaction costs of securities at the three periods are set as $d_{1,i} = 0.001$, $d_{2,i} = 0.002$, and $d_{3,i} = 0.003$, respectively. The reference return $\theta = W_0$. The probability of weighting function is set as $p = 0.5$. $\alpha = 0.15$ is the highest risk tolerance threshold for the portfolios. We suppose bankruptcy will occur when the wealth of the investor reaches zero, i.e., $\beta_t = 0$, $t = 1, 2, 3$, and set $\varepsilon_t = \{0.2, 0.2, 0.2\}$ as the level of insolvency belief for each of the three investment periods. Moreover, the following parameters are set for the solution algorithm: Population size is 20 and the maximum number of generations is 1000. The parameters $x^{min}$ and $x^{max}$ are set to 0 and 0.6, respectively.

Following the completion of one thousand cycles of the defined algorithm being applied to the model, and has gone through 20 independent experiments, we take their average value as our experimental results, and the respective investing methods are shown in Table 2, we can find that when the investor utilizes the model (25) to make his portfolio decision, he should follow the investment strategies listed in lines Table 2 to adjust his wealth at the beginning of each period. Namely, at the start of period 1, the investor needs to assign his initial wealth among Security 1, 2, 3, 4, 5, 6, 8, 9 and 10 by the investment proportions of 0.1532, 0.0089, 0.0725, 0.0810, 0.0832, 0.0995, 0.1254, 0.1398, 0.0781 and 0.1583, respectively. At the start of

**Table 2. Proportion of each security in the optimal portfolio.**

| Security $i$ | 1 | 2 | 3 | 4 | 5 | 6 | 7 | 8 | 9 | 10 |
|---|---|---|---|---|---|---|---|---|---|---|
| $t = 1$ | 0.1532 | 0.0089 | 0.0725 | 0.0810 | 0.0832 | 0.0995 | 0.1254 | 0.1398 | 0.0781 | 0.1583 |
| $t = 2$ | 0.1615 | 0.1038 | 0.1118 | 0.1439 | 0.0443 | 0.0935 | 0.0683 | 0.1108 | 0.0261 | 0.1361 |
| $t = 3$ | 0.0277 | 0.0000 | 0.0112 | 0.0404 | 0.4928 | 0.0360 | 0.0083 | 0.1533 | 0.2302 | 0.0000 |

**Table 3. Proportion of each security under different risk tolerance.**

| $\alpha$ | Utility | Optimal investment proportions |
|---|---|---|
| 0.10 | 0.0931 | $x_{1,i}$ = {0.1571, 0.0272, 0.0797, 0.0659, 0.0907, 0.0752, 0.0908, 0.1768, 0.0602, 0.1764} |
| | | $x_{2,i}$ = {0.1067, 0.0476, 0.0960, 0.0393, 0.1758, 0.1057, 0.0223, 0.1860, 0.0442, 0.1765} |
| | | $x_{3,i}$ = {0.1047, 0.0211, 0.1484, 0.0504, 0.2401, 0.0111, 0.0549, 0.0415, 0.1413, 0.1865} |
| 0.15 | 0.1065 | $x_{1,i}$ = {0.1532, 0.0089, 0.0725, 0.0810, 0.0832, 0.0995, 0.1254, 0.1398, 0.0781, 0.1583} |
| | | $x_{2,i}$ = {0.1615, 0.1038, 0.1118, 0.1439, 0.0443, 0.0935, 0.0683, 0.1108, 0.0261, 0.1361} |
| | | $x_{3,i}$ = {0.0341, 0.0000, 0.0138, 0.0513, 0.3733, 0.0573, 0.0102, 0.1885, 0.2714, 0.0000} |
| 0.20 | 0.1260 | $x_{1,i}$ = {0.0493, 0.0369, 0.0469, 0.0324, 0.1316, 0.0005, 0.1984, 0.1869, 0.1984, 0.1188} |
| | | $x_{2,i}$ = {0.0450, 0.0046, 0.0531, 0.0918, 0.0960, 0.0620, 0.1383, 0.1590, 0.1776, 0.1725} |
| | | $x_{3,i}$ = {0.0709, 0.0334, 0.0000, 0.0917, 0.0478, 0.0396, 0.0000, 0.5510, 0.1656, 0.0000} |

period 2, the investor must once again alter his fortune. Following modification, he holds Security 1, 2, 3, 4, 5, 6, 8, 9 and 10 by the investment proportions of 0.1615, 0.1038, 0.1118, 0.1439, 0.0443, 0.0935, 0.0683, 0.1108, 0.0261 and 0.1361, respectively. At the start of period 3, the investor must alter his fortune once again. During the investing period, he develops a portfolio consisting of Securities 1, 2, 3, 4, 5, 6, 8, 9 and 10 in the proportions 0.0277, 0, 0.0112, 0.0404, 0.4928, 0.0360, 0.0083, 0.1533, 0.2302 and 0.0, respectively. From Table 2 we can see that investor mainly invest his wealth in Security 5, 8 and 9, finally we get the crisp value of return utility is 0.1130.

To demonstrate the impact of cumulative risk and bankruptcy constraint on return utility, we use different risk tolerance and belief degree of bankruptcy to reflect return utility. Under each different case, 1000 generations of the developed algorithm indicate that the investor's capital should be allocated according to the investment methods stated in Tables 3 and 4.

Suppose the investor invests his or her money in accordance with optimum investment plan in Tables 3 and 4. From Table 3, when the cumulative risk is 0.10, the return utility is 0.0931, when the cumulative risk is 0.15, the return utility is 0.1065, and when the cumulative risk is 0.20, the return utility is 0.1260. In addition, under different cumulative risks, the optimal investment proportion of securities will change accordingly. For example, the optimal investment proportion of the first security will decrease with the increase of cumulative risks, and the optimal investment proportion of the seventh security will gradually increase with the increase of cumulative risks. Obviously, the bigger value of the cumulative risk $\alpha$, the bigger the value of the return utility will become. But often people are not willing to take a big risk, and they tend to a moderate risk value. Similarly, we derive from Table 4, when the belief

**Table 4. Proportion of each security under different belief degree of bankruptcy.**

| $\varepsilon_t$ | Utility | Optimal investment proportions |
|---|---|---|
| {0.1, 0.1, 0.1} | 0.0952 | $x_{1,i}$ = {0.1573, 0.1283, 0.0707, 0.0250, 0.1634, 0.1322, 0.10545, 0.1631, 0.0352, 0.0192} |
| | | $x_{2,i}$ = {0.2318, 0.0710, 0.0325, 0.0644, 0.0818, 0.0672, 0.0770, 0.2515, 0.0388, 0.0840} |
| | | $x_{3,i}$ = {0.1518, 0.0600, 0.0064, 0.0372, 0.2563, 0.0000, 0.1237, 0.2504, 0.1142, 0.0000} |
| {0.2, 0.2, 0.2} | 0.1229 | $x_{1,i}$ = {0.1573, 0.1283, 0.0707, 0.0250, 0.1634, 0.1322, 0.1055, 0.1631, 0.0352, 0.0192} |
| | | $x_{2,i}$ = {0.2318, 0.0710, 0.03245, 0.0644, 0.0818, 0.0672, 0.0770, 0.2515, 0.0388, 0.0840} |
| | | $x_{3,i}$ = {0.0157, 0.01994, 0.0000, 0.0000, 0.4979, 0.0000, 0.0734, 0.3202, 0.0729, 0.0000} |
| {0.3, 0.3, 0.3} | 0.1024 | $x_{1,i}$ = {0.1573, 0.1283, 0.0707, 0.0250, 0.1634, 0.1322, 0.1054, 0.1631, 0.0352, 0.0192} |
| | | $x_{2,i}$ = {0.2318, 0.0710, 0.0325, 0.0644, 0.0818, 0.0672, 0.0770, 0.2515, 0.0388, 0.0840} |
| | | $x_{3,i}$ = {0.1328, 0.0528, 0.0056, 0.0000, 0.2949, 0.0000, 0.1088, 0.3046, 0.1005, 0.0000} |

degree of bankruptcy is {0.1, 0.1, 0.1}, the return utility is 0.0952, when the belief degree of bankruptcy is {0.2, 0.2, 0.2}, the return utility is 0.1229, and when the belief degree of bankruptcy is {0.3, 0.3, 0.3}, the return utility is 0.1024. It can be seen that under different belief degree of bankruptcy, the results of the third period has changed, and the consequences of bankruptcy constraint and risk constraint on return utility are different. Not the bigger the bankruptcy value, the bigger the return utility. But different belief degree of bankruptcy also cause different return utility. Consequently, we may infer that both the cumulative risk and the bankruptcy restriction influence the optimum portfolio composition.

## Conclusion

In this study, we looked into an uncertain multi-period portfolio selection problem that took into account transaction costs and investor bankruptcies. The return utility and risk of an investment were determined by the utility function from prospect theory and the uncertain semivariance, respectively. The suggested model is a multiperiod programming problem whose aims are to maximize the terminal return utility. At the same time, minimize the value of cumulative risk and belief degree of bankruptcy over the whole investment horizon. In addition, a revised method for an artificial bee colony is devised to solve the stated programming challenge. A numerical example is provided to explain the concept of our model and show the efficacy of the algorithm's design. The calculation results demonstrate that the suggested model may convey investors' intentions by adjusting the satisfaction degree parameter values. In future work, we are able to apply the suggested approach to a hybrid portfolio selection issue using actual market data. In addition, the novel artificial bee colony technique suggested in this study may be used to various optimization issues, including vehicle scheduling and optimum route planning. Finally, we would choose assets using machine learning approaches [42, 43], then incorporate the selected assets into the portfolio selection model.

## Supporting information

**S1 Data.**
(XLS)

**S1 Dataset.**
(XLS)

## Author Contributions

**Formal analysis:** Gaohuizi Guo.

**Funding acquisition:** Cuiyou Yao.

**Methodology:** Gaohuizi Guo.

**Software:** Yao Xiao.

**Visualization:** Yao Xiao.

**Writing – original draft:** Gaohuizi Guo, Yao Xiao.

**Writing – review & editing:** Cuiyou Yao.

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
