## [Decision Letter · Decision Letter 0]

30 May 2022

PONE-D-22-06743Multi-period uncertain portfolio selection model with prospect utility functionPLOS ONE

Dear Dr. Guo,

Thank you for submitting your manuscript to PLOS ONE. After careful consideration, we feel that it has merit but does not fully meet PLOS ONE’s publication criteria as it currently stands. Therefore, we invite you to submit a revised version of the manuscript that addresses the points raised during the review process.

Please address all the comments from the reviewers and a careful proof reading is highly recommended for the revision.

We look forward to receiving your revised manuscript.

Kind regards,

Qichun Zhang, PhD

Academic Editor

PLOS ONE

Journal Requirements:

2. Thank you for submitting the above manuscript to PLOS ONE. During our internal evaluation of the manuscript, we found significant text overlap between your submission and the following previously published works, some of which you are an author.

- https://www.sciencedirect.com/science/article/abs/pii/S0377221712003165?via%3Dihub

- https://linkinghub.elsevier.com/retrieve/pii/S0020025512004197

- https://mountainscholar.org/bitstream/handle/10217/206716/Wang_TY_ExpSysApp_2015.pdf?isAllowed=y&sequence=1

- http://orsc.edu.cn/online/180223.pdf

- https://www.sciencedirect.com/science/article/abs/pii/S037722171400900X?via%3Dihub

Please revise the manuscript to rephrase the duplicated text, cite your sources, and provide details as to how the current manuscript advances on previous work. Please note that further consideration is dependent on the submission of a manuscript that addresses these concerns about the overlap in text with published work.

“This work was supported by Beijing Municipal Natural Science Foundation(No.9192005) and

the special fund of basic scientific research business fees of Beijing Municipal University of Capital

University of Economics and Business.”

“This work was supported by Beijing Municipal Natural Science Foundation(No.9192005) and the special fund of basic scientific research business fees of Beijing Municipal University of Capital University of Economics and Business.”

 “This work was supported by Beijing Municipal Natural Science Foundation(No.9192005) and

the special fund of basic scientific research business fees of Beijing Municipal University of Capital

University of Economics and Business.”

7. In your Data Availability statement, you have not specified where the minimal data set underlying the results described in your manuscript can be found. PLOS defines a study's minimal data set as the underlying data used to reach the conclusions drawn in the manuscript and any additional data required to replicate the reported study findings in their entirety. All PLOS journals require that the minimal data set be made fully available. For more information about our data policy, please see http://journals.plos.org/plosone/s/data-availability.

Additional Editor Comments:

The paper is well-organized with solid contribution. The results have been given clearly and sufficiently. All the reviewers' comments show that the paper is acceptable after a minor revision. In particular, some typos in the paper should be corrected and a proof reading is helpful to improve the quality of the presentation. The motivation and the background of the study should also be highlighted with more details where including more reference to support the motivation may be a proper approach. Therefore, a minor revision is needed to respond all the comments.

Reviewers' comments:

Reviewer's Responses to Questions

**Comments to the Author**

1. Is the manuscript technically sound, and do the data support the conclusions?

Reviewer #1: Yes

Reviewer #2: Yes

Reviewer #3: Yes

2. Has the statistical analysis been performed appropriately and rigorously? 

Reviewer #1: Yes

Reviewer #2: Yes

Reviewer #3: Yes

3. Have the authors made all data underlying the findings in their manuscript fully available?

Reviewer #1: Yes

Reviewer #2: Yes

Reviewer #3: Yes

4. Is the manuscript presented in an intelligible fashion and written in standard English?

Reviewer #1: Yes

Reviewer #2: Yes

Reviewer #3: Yes

5. Review Comments to the Author

Reviewer #1: This is a detailed paper that proposes a new improved artificial bee colony (ABC) method for multi-period uncertain portfolio selection. The explanation on how to improve it are well presented. However, when compared to other techniques, the choice of ABC is not insufficiently discussed. By introducing CPT for the first time, the abbreviations should be used correctly to improve the article.

Reviewer #2: This paper discusses a multi-period uncertain portfolio selection model with a prospect utility function and designs a new artificial bee colony algorithm by combining the sine cosine search method. The paper is overall enjoyable, and the originality of this paper is high. However, the reviewer still has the following comments which need to be addressed.

1. This paper is advised to revise the literature review section, including these related references. The authors can consider including the following research works to enrich the literature study.

(i)https://www.sciencedirect.com/science/article/abs/pii/S1568494615005955/

(ii) https://link.springer.com/article/10.1007/s12652-017-0478-4

2. This paper introduces the ABC algorithm and proposes a novel improved method. However, different algorithms have different applications. The author should introduce the application of the ABC algorithm in the field of the portfolio.

3.Some grammar errors should be noticed and updated as formal English writing style.

Reviewer #3: This manuscript describes an improved artifificial bee colony algorithms for a multi-period uncertain portfolio selection problem. Numerical results prove the effectiveness of the algorithm under reasonable assumptions, and data analysis is performed appropriately. It is interesting and organized well. The manuscript can be accepted after correcting grammartical、spelling errors.

For example:

1.After definition 3, in “provided that at least one of the two integrals is fifinite.” here, “provided” should be “Provided”.

2.In equation (26) has been given a comma symbol, it should be given a full stop symbol.

3.In line 6, "makes an one-off... " is incorrect.

4.In line 12, "programing" should be “programming”.

5.In the line 10 of second paragraph, the spaces of "returns . For example ," should be deleted.

6. PLOS authors have the option to publish the peer review history of their article (what does this mean?). If published, this will include your full peer review and any attached files.

Reviewer #1: No

Reviewer #2: No

Reviewer #3: No

---

## [Author Response · Author response to Decision Letter 0]

20 Aug 2022

Dear Editor, 

We greatly appreciate the editor’s and reviewers’ insightful review of our manuscript entitled “Multi-period uncertain portfolio selection model with prospect utility function”. The comments have helped us make substantial improvements to this manuscript. Following the Editor’s recommendations along with Referees comments, we have revised the manuscript carefully. The main revisions have been marked in red in the revised manuscript. In addition, we did our best to revise the manuscript and rephrased the duplicate text. 

We sincerely hope that contents of this paper in present form would be useful for readers of the PLOS ONE, and revised manuscript will be acceptable to you, Editor and the esteemed referees. This manuscript represents the authors’ original work and has not been published nor has it been submitted simultaneously elsewhere. All authors have checked the manuscript and have agreed to the submission. 

Waiting for your early and positive response. 

Thanking you 

Dr. Gaohuizi Guo 

Corresponding Author

Respond to the Reviewer 1 Comments

Comment 1: This is a detailed paper that proposes a new improved artificial bee colony (ABC) method for multi-period uncertain portfolio selection. The explanation on how to improve it are well presented. However, when compared to other techniques, the choice of ABC is not insufficiently discussed. By introducing CPT for the first time, the abbreviations should be used correctly to improve the article.

Reply: Thanks for your suggestions. According to your valuable comments, we have 

revised the manuscript carefully. Firstly, we have corrected the abbreviation CPT, which was first introduced on page 8. Second, we compare the ABC algorithm with other techniques on page 9 and explain why we chose the ABC algorithm.

Respond to the Reviewer 2 Comments

Comment 1: This paper is advised to revise the literature review section, including these related references. The authors can consider including the following research works to enrich the literature study.

(i)https://www.sciencedirect.com/science/article/abs/pii/S1568494615005955/

(ii) https://link.springer.com/article/10.1007/s12652-017-0478-4

Reply: Thanks for your suggestions. We have revised the literature review section, added newer references, and enriched the study of the literature. These include the study of solving multi-period portfolio problems and the study of multi-period portfolio optimization problems in uncertain environments. They are marked in red in the text.

Comment 2: This paper introduces the ABC algorithm and proposes a novel improved method. However, different algorithms have different applications. The author should introduce the application of the ABC algorithm in the field of the portfolio.

Reply: Thanks for your suggestions. We introduce the application of the ABC algorithm in the field of the portfolio on page 10 and introduce the relevant references.

Respond to the Reviewer 3 Comments

Comment 1: This manuscript describes an improved artifificial bee colony algorithms for a multi-period uncertain portfolio selection problem. Numerical results prove the effectiveness of the algorithm under reasonable assumptions, and data analysis is performed appropriately. It is interesting and organized well. The manuscript can be accepted after correcting grammartical、spelling errors.

Reply: Thanks for your suggestions. According to your valuable comments, we carefully checked the entire article for grammatical problems and corrected them one by one, and marked them all in red in the text.

---

## [Decision Letter · Decision Letter 1]

1 Sep 2022

Multi-period uncertain portfolio selection model with prospect utility function

PONE-D-22-06743R1

Dear Dr. Guo,

We’re pleased to inform you that your manuscript has been judged scientifically suitable for publication and will be formally accepted for publication once it meets all outstanding technical requirements.

Kind regards,

Qichun Zhang, PhD

Academic Editor

PLOS ONE

Additional Editor Comments :

Since all the concerns have been addressed well in the revised version and no further comments are received, I recommend accepting this manuscript as it is.

Reviewers' comments:

Reviewer's Responses to Questions

**Comments to the Author**

1. If the authors have adequately addressed your comments raised in a previous round of review and you feel that this manuscript is now acceptable for publication, you may indicate that here to bypass the “Comments to the Author” section, enter your conflict of interest statement in the “Confidential to Editor” section, and submit your "Accept" recommendation.

Reviewer #2: (No Response)

Reviewer #3: All comments have been addressed

2. Is the manuscript technically sound, and do the data support the conclusions?

Reviewer #2: Yes

Reviewer #3: Yes

3. Has the statistical analysis been performed appropriately and rigorously? 

Reviewer #2: Yes

Reviewer #3: Yes

4. Have the authors made all data underlying the findings in their manuscript fully available?

Reviewer #2: Yes

Reviewer #3: Yes

5. Is the manuscript presented in an intelligible fashion and written in standard English?

Reviewer #2: Yes

Reviewer #3: Yes

6. Review Comments to the Author

Reviewer #2: The authors have done some suitable updates in the revised manuscript to correct the mentioned errors. The author's response to reviewer comments has proved their work fits the criteria for publication.

Reviewer #3: (No Response)

7. PLOS authors have the option to publish the peer review history of their article (what does this mean?). If published, this will include your full peer review and any attached files.

Reviewer #2: No

Reviewer #3: No

---

## [Editor Report · Acceptance letter]

5 Sep 2022

PONE-D-22-06743R1 

Multi-period uncertain portfolio selection model with prospect utility function 

Dear Dr. Guo:

I'm pleased to inform you that your manuscript has been deemed suitable for publication in PLOS ONE. Congratulations! Your manuscript is now with our production department. 

Kind regards, 

on behalf of

Dr. Qichun Zhang 

Academic Editor

PLOS ONE